# Canadian Greenhouse Operations and Their Potential to Enhance Domestic Food Security

Gabriel LaPlante [1,†], Sonja Andrekovic [1,†], Robert G. Young [2,*], Jocelyn M. Kelly [2], Niki Bennett [3], Elliott J. Currie [4] and Robert H. Hanner [2]

1   Department of Molecular and Cellular Biology, University of Guelph, 50 Stone Rd E,
    Guelph, ON N1G 2W1, Canada; glaplant@uoguelph.ca (G.L.); sandreko@uoguelph.ca (S.A.)
2   Department of Integrative Biology, University of Guelph, 50 Stone Rd E, Guelph, ON N1G 2W1, Canada;
    jkelly17@uoguelph.ca (J.M.K.); rhanner@uoguelph.ca (R.H.H.)
3   Ontario Greenhouse Vegetable Growers, 32 Seneca Rd, Leamington, ON N8H 5H7, Canada;
    nbennett@ontariogreenhouse.com
4   Gordon S. Lang School of Business and Economics, University of Guelph, 50 Stone Rd E,
    Guelph, ON N1G 2W1, Canada; ecurrie@uoguelph.ca
*   Correspondence: rgyoung6@gmail.com
†   Both authors equally contributed to this manuscript.

**Abstract:** Food security is a growing societal challenge. The pressure to feed a projected global population of 9.6 billion by 2050 will continue to be limited by decreasing arable land. The recent disruptions in international trade resulting from responses to the COVID-19 pandemic have highlighted the importance of regional self-reliance in food production. While Canada is highly self-reliant in food categories such as meat and dairy, the nation relies heavily on international imports to fulfill fresh vegetable demands. In potential future scenarios where international trade faces disruptions, Canadian food security could be at risk. By providing local sources of fresh foods year-round, the greenhouse vegetable industry holds strong potential to overcome future food supply shortages and could become a critical contributor to self-sustainable food production in Canada. Many challenges, however, surround the Canadian greenhouse industry. Some challenges include the persistence and spread of infectious plant pathogens and forecasted labour shortages. Opportunities to alleviate such challenges include introducing more diverse commodity groups and integrating innovative technologies to accelerate efficiency within the industry. In this commentary, we examine the current state of the Canadian greenhouse industry, explore potential challenges, and highlight opportunities that could promote food security across the nation.

**Keywords:** food security; greenhouse; COVID-19; biosurveillance



## 1. Introduction

With the global population predicted to reach 9.6 billion by 2050, feeding the world will become an increasingly difficult and complex challenge [1] (pp. 483–498), [2] (pp. 902–905). It is forecasted that climate change will continue to threaten agricultural productivity with the occurrence of more frequent and extreme weather patterns [3,4] (pp. 306–321). Additionally, the emergence of the COVID-19 pandemic has highlighted the importance of creating sustainable self-reliant food economies [5] (pp. 871–876). The pandemic has exacerbated global food insecurity, with predictions nearly doubling the number of food-insecure individuals from 135 million to 265 million between January and December of 2020 [6]. Even in nations that are currently considered food secure, like Canada, food supply could be threatened if foreign trade cannot be relied upon to fulfill consumer demands. One of the primary challenges for countries with shorter growing seasons is providing fresh vegetable produce domestically year-round. In Canada, the vegetable greenhouse industry may present an option for potential solutions.

The global responses to the COVID-19 pandemic have placed pressures on national food supply chains [5] (pp. 871–876), [7]. Border closures and trade halts have put direct pressure on domestic food supplied by international markets while also limiting the migration of essential seasonal labourers [7,8] (pp. 163–169). Canada relies heavily on international trade, importing 3.6 billion Canadian Dollars (CAD) worth of fresh vegetables in 2018 [9]. To protect national food security, the development of year-round vegetable production through greenhouse operations in Canada could support a more robust and resilient supply chain [8] (pp. 163–169).

Despite its significant contribution to Canada's agri-food industry, the Canadian greenhouse industry can still become a stronger component of a national food system, especially considering the impacts of COVID-19 [10] (pp. 219–224). Increased operating expenses, decreased labour availability, yield losses attributed to plant pests and pathogens, and lack of publicly available data are some areas identified in this commentary as prohibitive factors to the industry's expansion. To overcome these challenges, Canada must leverage opportunities to increase greenhouse productivity and efficiency. Expanding vegetable production and diversifying commodity groups could enable the industry to better meet Canadian demands. By strategically selecting regions for new greenhouse development, as well as through the application of innovative technologies, the industry could benefit from increased productivity and higher yields. In turn, this would require increased investment in education and innovation to facilitate the long-term growth of the greenhouse industry and work toward furthering Canadian food security. If the identified challenges are addressed and opportunities leveraged, the greenhouse industry could alleviate some of the food security concerns in Canada.

## 2. Current State of the Canadian Vegetable Greenhouse Industry

### 2.1. Size, Structure, and Trends

The greenhouse vegetable sector is both the largest and fastest growing area of Canadian horticulture [9,11]. Common vegetables grown in Canadian greenhouses include tomatoes, cucumbers, lettuce, peppers, green beans, eggplant, herbs, and microgreens; however, tomatoes, cucumbers, and peppers represent an overwhelming majority [11], [12] (pp. 23–28) [13,14]. Collectively, tomatoes, cucumbers, and peppers represent 96% of the total harvested area (16.8 million square meters), 98% of production by metric tonne, and 95% of total farm gate value of all Canadian greenhouse vegetables [9].

Since 2011, the Canadian vegetable greenhouse industry has experienced annual upward trends in growth [13]. For example, the province of Ontario had an increase of 30% in total harvested area between 2011 and 2016 [15]. This growth is largely attributed to trends in local food demand by consumers and influences of international markets [13,15–17]. Like harvested area, Canada-wide sales of greenhouse vegetables have also increased. Between 2017 and 2018, sales increased by 5.2% to a value of 1.5 billion CAD, with a total greenhouse area reaching 26.1 million square meters [18]. Ontario continues to lead the Canadian greenhouse vegetable sector, accounting for 68% of total operations, 70% of total harvested area, and 960 million CAD in farm gate value in 2018 [9]. The continued expansion of the Canadian vegetable greenhouse industry provides an opportunity for Canada to increase its domestic food supply, especially in times of unforeseen border closures where a large portion of Canada's food supply could be halted. If the Canadian greenhouse industry is to expand to better sustain Canada's domestic food needs, a multifaceted approach capitalizing on existing strengths while addressing gaps in our produce supply is needed. The Canadian greenhouse industry has the potential to increase production of current top-producing crops and diversify to include other crops for which domestic markets exist. Existing greenhouse operations in highly productive regions can increase efficiency and productivity with the incorporation of novel agricultural technologies. Finally, we see the need for expanding operations to non-traditional locations, such as urban and remote Canadian locations.

*2.2. International Trade*

Despite the strength of Canada's vegetable sector, it still relies heavily on international trade to fulfil its fresh vegetable needs. The most recent available data reported vegetable imports from international markets at quantities of 1.99 million metric tonnes, or values of approximately 3.6 billion CAD annually [9]. The majority of Canada's vegetable imports are consistently supplied by the USA and Mexico [11,19]. In 2018, the two countries provided 87.8% of the total vegetable imports, with 60.1% (approximately 2.2 billion CAD) of imports from the USA and 27.7% (approximately 1 billion CAD) from Mexico [9]. The top vegetables imported from the USA by Canadian dollar are cabbage, lettuce, and broccoli/cauliflower, while the top imports from Mexico include peppers, tomatoes, and cucumbers [9]. Too much reliance on food imports leaves nations vulnerable to price fluctuations and shortages [20] (pp. 175–201) [21] (pp. 398–410) [22]. As Canada relies on international imports to fulfil the fresh vegetable demand, the recent disruptions in trade with the USA and Mexico have challenged Canada's ability to remain food secure. If properly leveraged, the greenhouse industry has the capacity to supply more vegetables and alleviate some of the demand for international produce, enhancing Canadian food security.

Canada also exports a significant number of vegetables internationally [23] (pp. 424–457). In 2018, over half (51.2%) of the total fresh vegetable exports from Canada were grown by greenhouse operations, contributing approximately 1.95 billion CAD to the Canadian economy [9,11]. As well as being Canada's largest supplier of vegetables, the USA remains Canada's largest importer. In 2018, the USA imported 97.8% of Canada's total vegetable exports, amounting to values of approximately 1.91 billion CAD [9]. The Canadian approach to international food trade, specifically horticultural products, has been criticized for its inefficient reliance on imports that could be supplied domestically [23] (pp. 424–457). While some of the disparity can be attributed to the seasonal nature of vegetable and fruit production, more of Canada's food could be supplied domestically if it were better encouraged by agricultural and economic policy [23] (pp. 424–457) [24]. Given the current national dependence on produce imports, and conversely, the dependence of the Canadian greenhouse industry economy on exporting its top crops, the development of a more robust domestic supply chain facilitated by greenhouse industry expansion is needed. While we recognize the many benefits of globalization and international agricultural trade [25], focusing on domestic supply could strengthen food security in future scenarios where maintaining the current level of trade is not feasible.

*2.3. Challenges Facing the Canadian Greenhouse Industry*

The vegetable greenhouse industry must overcome unique challenges to expand. Increasing operating expenses [18], economic losses due to pest incidence [26,27] (pp. 35–42), labour shortages [28,29], wasted crops [30], and a lack of publicly available industry data are explored here as they relate to the productivity and sustainability of the Canadian greenhouse industry.

Increased operating expenses are a common obstacle experienced by the Canadian greenhouse industry. Operating expenses for greenhouse growers increased by 1.6% in 2018 and by 5% in 2019, and the largest share of greenhouse expenses is currently attributed to labour costs, accounting for 31% in 2019 [11]. Another area contributing to increased costs is integrated pest management [18]. Labour- and time-intensive crop protection practices, such as pesticide applications, biological control agents (BCAs), and sanitization and maintenance procedures, are necessary to limit crop destruction by pests and pathogens [18,31,32]. Insects such as pepper weevils, cucumber beetles, thrips, and whiteflies can all pose serious economic threats through greenhouse crop destruction [33]. For example, the Ontario Greenhouse Vegetable Growers (OGVG) reported provincial losses of up to 67 million CAD resulting from a pepper weevil population boom in 2016 [34]. Bacterial, viral, and fungal pathogens also contribute to yield losses; common species include *Pseudoperonospora cubensis*, *Fusarium solani*, *Fusarium oxysporum*, and *Pepino mosaic* [33]. In 2017, an *F. oxysporum* fungal outbreak infected pepper greenhouses in Essex County, Ontario, resulting in major plant loss; in a single

example, one greenhouse operation experienced over 12% pepper mortality [35] (pp. 121–132). One challenge with many crop pathogens is their largely undetectable presence at early stages of infection, allowing mitigation only after significant crop damage has occurred [36] (pp. 2887–2904). For this reason, visual inspections alone for pathogen monitoring may not be adequate, as even the most rigorous inspections are unlikely to detect infection in a timely manner for successful mitigation efforts [36] (pp. 2887–2904), [37] (pp. 1351–1457). The implementation of technologies such as automation and molecular biosurveillance tools could help mitigate high operating costs in the future and support the greenhouse industry's expansion.

Labour shortages present a long-standing challenge for Canadian vegetable agriculture in both greenhouse and field operations, though greenhouses' labour needs are higher, partly due to their longer growing season [1] (pp. 483–498), [38]. In the last 20 years, the agricultural sector experienced a 31% decline in its workforce and 37% of the current workforce is forecasted to retire in the next decade [39]. This could account for nearly 114,000 job vacancies and could result in losses of over 1.5 billion CAD annually if positions are left unfilled [28,39]. The challenge of labour shortages can be partially attributed to shifting demographics within the agricultural sector as the rate of retiring agricultural workers increases [28]. In addition, the greenhouse industry relies heavily on the seasonal recruitment of employees, especially temporary foreign workers, to sustain operations. Labour sourcing has become even more challenging due to international lockdown measures resulting from COVID-19 [40]. Travel restrictions preventing international worker flow into Canada has resulted in many unfilled agricultural positions, leading to reported crop losses of over 50% in some field operations [29]. Labour shortages in the USA have left many fields unharvested in one of Canada's major trading partners [30]. According to the National Sustainable Agriculture Coalition report, between March and May 2020, the USA lost an estimated 1.32 billion USD (1.77 billion CAD) from wasted field vegetables [30]. These losses directly affect Canada's fresh vegetable supply and may put further pressure on the greenhouse industry to fill the supply gap. Supporting the Canadian greenhouse industry to be better prepared for impacts on the national food supply, especially in cold seasons, is prudent given recent global pressures. If greenhouse operation expansion is a future goal, addressing labour shortages must be a priority. This issue should be approached from two sides: decreasing the need for human labour through automation and efficiency increases, and increasing the human workforce through enhanced recruitment, education, and training [41–44].

Another major challenge facing the greenhouse industry is the lack of publicly available industry data. For example, pathogen-associated crop losses pose a major threat to greenhouse operators in Canada, but quantitative justification for implementing preventative and mitigative strategies is difficult to obtain due to the lack of centralized reported and available data. Access to accurate and reliably updated data would provide transparency on pathogen-associated losses in the greenhouse industry, thus providing more opportunities to implement new and more effective management techniques. Some research-sharing networks already exist for monitoring plant pathogen threats, such as the Better Border Biosecurity (B3) in New Zealand and the Plant Biosecurity Research Initiative (PBRI) in Australia, but a need for increased local and global information connectivity has been identified [2] (pp. 902–905) [45] (pp. 1237–1239). Furthermore, industry data, such as different commodity values and target market information, are not readily available. This may be an obstacle for the entry of new growers into the industry, potentially inhibiting expansion of greenhouse operations [46–48].

## 3. Opportunities for the Canadian Vegetable Greenhouse Industry

The Canadian greenhouse industry has been steadily experiencing upward trends in growth over the past decade; however, many opportunities exist for further expansion and for meeting industry challenges. Below, we discuss diversification of commodity groups, regional expansion of greenhouse operations, implementation of innovative technology, and prioritization of environmental sustainability as key opportunities. By focusing on the development of the aforementioned areas to improve greenhouse efficiency, Canada's

capacity for generating a self-reliant food supply can be enhanced, thereby increasing national food security.

### 3.1. Increasing Canada's Self-Reliance by Expanding Commodity Groups

Crop choices are largely driven by trade and economic forces, which favour specialization over diversification [49,50] (pp. 222–231). The socio-economic advantages of globalization, agricultural specialization, and economies of scale are well-recognized [25,51,52]. However, crop diversification provides resilience against volatile conditions that can impact food security, such as uncertain trade environments, pathogen spread, or climate events [53] (pp. 183–193) [54–57] (pp. 795–808). Many countries, especially in northern Europe, have implemented a more diverse domestic crop portfolio in an effort to reduce water trade and increase self-reliance [58]. In regions with harsh year-round growing environments and lower annual temperatures, such as Canada, the greenhouse industry is well-suited to lead crop diversification to expand its profile of fruits and vegetables and enhance food security. Furthermore, the greenhouse industry is more amenable to crop diversification than other forms of horticulture, and the benefits are often more readily recognized by growers [59] (pp. 105–120).

The production of more diverse greenhouse vegetables could promote self-reliance in Canadian food supply chains. More specifically, the addition of non-traditional produce, including various ethno-cultural vegetables, could be explored. The rising demand for such commodities, such as okra and long beans, especially within Chinese, South Asian, and Afro-Caribbean communities, has been recognized in recent years as a significant market opportunity [60–64]. In 2010, the total market size for ethno-cultural vegetables in the Greater Toronto Area was estimated at 732 million CAD per year [60]. While some new commodities may require research investment for the optimization of their indoor growth, many ethno-cultural vegetables are already well-suited for greenhouses due to their high temperature and moisture requirements [65,66]. Additionally, increasing the production capacity of commodities already grown in greenhouses, such as lettuce, beans, eggplants, strawberries, and microgreens, could reduce Canada's trade reliance and promote a self-sustaining vegetable supply [67].

### 3.2. Regional Expansion of Canada's Vegetable Greenhouse Industry

Currently, the majority of the nation's vegetable greenhouses operate in Southern Ontario, with the Windsor-Essex County responsible for 60% of operations [9,68]. Expanding vegetable greenhouse operations to more remote, rural, and urban locations in Canada could potentially alleviate distribution inefficiencies and provide consumers with more affordable and accessible fresh food options year-round [1] (pp. 438–498) [69,70].

Despite Canada's relative food security, the issue of food affordability is a challenge for many families. This issue is especially prevalent in northern Canada; in 2019, it was estimated that 72% of children in the northern regions did not have reliable access to nutritious or affordable food [71]. The infrequent and weather-dependent shipments of fresh vegetables to these areas inflate costs to the point where making healthy choices becomes an unrealistic option [72]. Across the nation, as individuals continue to seek more nutritious diets, partly due to the new Canada's Food Guide promoting increased vegetable consumption, accessibility to healthy and fresh produce is becoming increasingly important [73]. However, vegetable prices increased 17% in Canada from 2018 to 2019, and many consumers lack the time and money to follow the 2019 Canadian food guide which encourages plant-based meals [74,75]. Even before the COVID-19 pandemic, reports predicted that increased protectionism, global growth slowdowns, and climate change will continue to result in increased food prices across Canada [74].

The development of greenhouse operations closer to consumer centers could result in decreased distribution costs, thus increasing food security to Canadians [72]. Research is active in Canada looking at the various implications of northern greenhouse operations [76] (pp. 31–38) [77,78]. In these northern regions, harsh weather conditions pose a challenge for

greenhouse development, but deep-winter greenhouses do exist [72]. Green Iglu (formerly Project Growing North), one of the first companies to construct winterized greenhouses in Nunavut, created a 1300 square foot polycarbonate structure that retained heat from a local water supply and was able to withstand seven feet of snow as well as winds reaching 170 km/h [72,79]. This winterized greenhouse has already provided a communal growing space for local residents, where they are able to grow and purchase fresh produce at a fraction of the price of imported foods [71]. In Atlantic Canada, a geothermal greenhouse has been implemented to produce vegetables for the Potlotek First Nation community on Cape Breton Island [80]. Further innovation and incorporation of greenhouses to the food supply chains of northern and remote Canadian communities can take economic strain off residents while enabling more reliable access to healthy foods. Additional benefits would include creating Canadian jobs, stimulating local economies, and further bolstering the Canadian greenhouse industry.

As agricultural land is strained under rising global populations, urban horticulture could represent a critical component for Canadian food security. In countries with colder climates, urban horticulture systems provide promise for additional sources of produce. Furthermore, urban agriculture can provide an environmentally sustainable alternative to conventional agriculture [81] with a significant decrease in water requirements as compared to conventional methods, the creation of fewer carbon emissions due to reduced transport in shipping product, and the overall use of less chemicals for preservation during long transport periods [81]. Gotham Greens, a rooftop greenhouse company with multiple locations in New York City and Chicago, claims that they produce vegetables at up to 30 times the productivity rates of a traditional operation, while their hydroponic systems reduce water usage by 95% [82]. Another major advantage of urban horticulture is maximizing the efficiency of land and resource use. Rooftop greenhouses around the world boast the benefits of preserving arable land while saving energy, utilizing excess heat from the building below or nearby industrial operations [82]. Lufa Farms is a Montreal-based business that aims to feed urban residents more sustainably, delivering greenhouse-grown vegetables via a subscription service. Their website asserts that they use half the heating energy that would be required for a comparable ground-level operation. Studies have indicated that it could be possible for some cities to achieve complete self-reliance in produce, contributing to food security while minimizing environmental impacts [83] (pp. 1–11). For example, Grewal and Grewal (2012) reported that Cleveland, Ohio, has the potential to meet the vegetable requirements of its 400,000 residents autonomously if hydroponic greenhouses were developed on every available rooftop [83]. While this would be a massive undertaking, achieving the same crop yields with conventional agricultural methods would take 14 times more space. Impacts from economic crises, such as those stemming from COVID-19, can create opportunities for greenhouse expansion on newly vacant land due to economic downturn that provide space for new or expanding operations. [1] (pp. 483–498). Further, a surplus of greenhouse space in British Columbia and Ontario exists due to the volatility of the Canadian cannabis industry, which could potentially be utilized to bolster greenhouse vegetable production [66]. The incorporation of community-centred greenhouse operations can serve urban residents while taking pressure off rural agricultural lands and balancing food availability between urban and rural areas [84] (pp. 33–51).

### 3.3. Technological Innovations to Improve Canadian Greenhouse Operations

Applying innovative technology, including molecular biosurveillance tools, data networks, and improvements in environmental sustainability, will be of paramount importance to increase the productivity of Canadian greenhouse operations. The utilization of existing and emerging technology could provide efficiencies that would enable the expansion of the industry on a national scale.

Management of plant pathogens is a critical practice for greenhouse operators given the propensity of greenhouse environments to harbor and spread pathogens rapidly. Best practices and pest regulations for managing pathogens are provided in Canada by national

and provincial organizations, such as the Ontario Ministry of Agriculture Food and Rural Affairs (OMAFRA) and the Canadian Food Inspection Agency (CFIA) [85,86]. Current surveillance recommendations, however, which include insect scouting and weekly visual inspections for signs of pathogen infection [87], may not be adequate in addressing visually undetectable threats. Current techniques can also result in struggling to distinguish between similar diseases and detecting early and asymptomatic stages of pathogen infection [88] (pp. 282–296). Presently, the only option for greenhouse operators to obtain specific identification of pests or pathogens is to send samples for costly lab testing and wait days to weeks for results [89]. Thus, a more proactive pathogen surveillance approach is critical to reduce future crop losses [90] (pp. 290–302).

The ability of molecular biosurveillance tools to provide specific, sensitive, and early plant pathogen detection could protect greenhouse operations from crop losses if used regularly. Important factors for the uptake of surveillance tools by greenhouse operators are sensitivity, cost-effectiveness, and ease of use. Handheld portable devices that utilize DNA-based probes are fast, accurate, and increasingly affordable, making them a promising avenue for the greenhouse industry [91] (pp. 250–258) [92] (pp. 499–505) [93] (pp. 5563–5568) [94] (pp. 2453–2468). For example, handheld polymerase chain reaction (PCR) thermocyclers have been commercialized for biosurveillance use, which can connect to mobile devices and generate results in 30–60 min [95] (pp. 2474–2479). Environmental DNA (eDNA) metabarcoding protocols have also recently been optimized for rapid point-of-use biosurveillance of pests and invasive species in Ontario forests and at Canada–US border points of entry [96–98] (pp. 1999–2014). Metabarcoding, a high-throughput-sequencing (HTS)-based method, has pros and cons compared to PCR-based biosurveillance, and the choice of tool implemented could depend on the specific application. For example, biosurveillance by PCR requires the generation of assays specific for the detection of a certain target organism, while metabarcoding will reveal multiple taxa present in a sample [94] (pp. 2453–2468). The cost of the testing can also vary greatly, as can the expertise needed to evaluate the resulting data, which is high for metabarcoding protocols. In either case, the implementation of a molecular biosurveillance strategy in greenhouses could be used to facilitate detection of fungal, bacterial, viral, or pest pathogens from plant samples before presentation of any visual symptoms, allowing early detection and minimizing crop losses. Adopting such technologies would decrease risks of yield loss to pathogens and enhance Canadian food security. A molecular biosurveillance pipeline exists for forest environments to detect invasive pest movement (Biosurveillance of Alien Forest Enemies), which could be used as a model for the development of a similar initiative in the Canadian agricultural sector, specifically the greenhouse industry [99] (pp. 95–115).

Another technological innovation to the greenhouse industry could be the integration of connected data networks for tracking pathogen diagnostics. Some international, national, and regional plant disease diagnostic networks already exist to enable efficient and accessible information sharing [2] (pp. 902–905) [100] (pp. 15–38). The need for connecting local, global, and national pathogen data has been identified and the generation of a Canadian network could increase connectivity between researchers, end-users, and collaborators to enhance productivity in the greenhouse industry [2] (pp. 902–905) [45] (pp. 1237–1239). As molecular surveillance techniques become more accessible, both technologically and economically, resources to support their integration into surveillance programs, such as known datasets to identify species or build PCR targeted tests, are needed [96]. It has been noted that it is the responsibility of governments to contribute to the public good by protecting both native and agricultural horticulture from pathogens [101]. While government funding supporting agricultural research, innovation, and knowledge translation exists in Canada [46–48], further support to initiate comprehensive data-sharing networks is needed and the recent efforts in Canada to establish a clean plant network, similar in concept to the United States of America National Clean Plant Network, are promising [2] (pp. 902–905) [102,103]. The development of a standardized diagnostic reporting network should be viewed as a pillar to maintaining optimal plant system health [100] (pp. 15–38). If such a system were to be coupled to a surveillance program using the molecular tools,

the greenhouse industry would benefit, and these developments could also spawn an associated industry with training, tools, and businesses supporting growers.

Technological innovation can also be applied to greenhouses to enhance environmental sustainability and aid the long-term success of the industry. The Ontario Greenhouse Vegetable Growers (OGVG) (representing all tomato, pepper, and cucumber growers in Ontario) encourages reducing resource use while generating maximum yield through the use of hydroponic systems [104]. A study by Zhou et al. (2021) highlighted that conventional, high-tech greenhouse operations scored the highest in four out of seven Sustainable Development Goals (SDGs) when compared to low-tech, conventional, and organic methods [105]. However, it was difficult for high-tech conventional greenhouses to achieve SDG7, involving clean and affordable energy [105]. Currently, energy costs comprise around 60% of total greenhouse operating expenses [106], but the adoption of innovative technologies could decrease costs while also improving environmental sustainability. In the Netherlands and Germany, energy saving materials, thermal screens, and solar energy storage in greenhouses were among the tools employed to reduce energy consumption by 80–90% [107] (pp. 271–277). It has been widely reported that the use of soilless growing materials, such as stone wool, greatly contribute to higher resource use efficiency and higher yields [103,108] (pp. 781–791) [109] (pp. 130–136) [110] (pp. 831–839). Determining the style of greenhouse is an important factor to consider when balancing energy costs. Dutch Venlo greenhouse structures, most commonly used in the Netherlands and Australia, maximize carbon dioxide enrichment, utilize efficient cooling and ventilation systems, and use horizontal screens to reduce heat demand and energy costs [105,111–113]. As well, using recirculating fertigation systems can contribute to recycling materials and reduced resource use [105]. Adopting intelligent automation, self-sufficient watering systems, and optimizing light levels are other technologies that can harness a smaller environmental footprint for greenhouse operations [114]. The large-scale implementation of energy-saving tools has benefited, and will continue to benefit, the Canadian greenhouse industry, while also aligning well with Canada's climate action plan [18,115].

Leveraging the key areas of technological innovation outlined above will help the greenhouse industry enhance Canada's self-sustainable food production, but this is by no means an exhaustive list. For example, another area of innovation that will aid the expansion of the greenhouse industry is furthering greenhouse use of automation and artificial intelligence [116]. One example of the use of artificial intelligence and automation is the use of climate-controlled algorithms to read environmental feedback, both inside and outside greenhouses, to respond quickly and accurately to changing conditions and provide consistent internal greenhouse growing conditions [116]. Increased investment in research and development and the recruitment of people into agri-food careers with emphasis on the implementation of new technologies can only enhance current greenhouse production. The human capital requirements accompanying continued development and application of technological innovation cannot be overlooked, and Canada is well-prepared with the educational infrastructure and government support to facilitate effective recruitment to support innovation.

### 3.4. Recruitment and Education of a Modern Agricultural Workforce

To support the adoption of technology and the expansion of the Canadian greenhouse industry, the promotion of the industry as a career option is a critical element. Increasing knowledge through promoting agricultural careers in curriculums at all levels is necessary. In a recent report, 84% of respondents aged 14–35 said they knew what career they wanted to pursue, and only 1% of these selected agriculture or natural resources [117]. More initiatives directed at engaging youth in agriculture would go a long way [118], and particular emphasis needs to be placed on raising awareness of agriculture as a viable career option for those from non-traditional agricultural regions (such as urban centers) and non-traditional agricultural backgrounds (such as computer programming, molecular biology, or mechanical engineering). The creation of a diversified and "modern" agricultural workforce is important for the future of the industry using technology, automation, and big

data [39]. Canada is poised to meet this challenge—with six Canadian educational institutions ranking among the top 100 agricultural programs globally, developing a dynamic, multi-faceted, interdisciplinary, and modern agricultural workforce is feasible [39].

## 4. Global Greenhouse Development Initiatives

The global community has been, and will continue to be, impacted by the effects of both climate change and COVID-19 on food security. Many countries have recently put a heightened focus on domestic supply, including greenhouse vegetable cultivation [119,120]. In Singapore, for example, border closures spurred efforts to realize the country's "30 by 30" goal of 30% of food needs met domestically by 2030, resulting in increased investment in high-tech greenhouse development [120].

There are added challenges for more northern countries with harsher climates, such as Canada, to increase greenhouse vegetable production. Finland, Sweden, Norway, Iceland, and Russia are looking to improve upon their existing greenhouse industries, which, like Canada, also commonly grow cucumbers, tomatoes, and peppers [121–124]. In fact, the Russian cucumber market is already almost 100% supplied by domestic greenhouse production [125]. In these northern regions, the major challenge for greenhouse production is the high energy requirements for year-round heating [126]. Innovative solutions are continually being developed. In Sweden, over the past decade, wood chip boilers have largely replaced fossil fuels to become the major greenhouse heating method [122]. Some operations use waste heat from nearby factories or biogas generators from adjacent farms, and implementation of confluent air jet climate control systems is being explored commercially [122]. Research has also indicated the potential for heating greenhouses using manure-based biogas where it is expected that up to 11% of the Swiss tomato demand could be supplied domestically in this manner [127]. Icelandic greenhouse operations have been reported to harness geothermal heat to extend their growing season [123].

In urban areas, the problem of heating can be addressed by developing rooftop greenhouses coupled to the buildings below or nearby. In a dually beneficial manner, rooftop greenhouses provide insulation that prevents heat loss in the winter and absorbs excess heat in the summer, preventing detrimental urban heat island effects [82]. In addition to heat waste, urban greenhouses can be coupled to industrial or residential buildings to make use of carbon dioxide outputs [82,128]. Countries across the Americas and Europe have developed large-scale rooftop greenhouses [82,128–130], and an analysis has indicated that, in addition to these larger efforts, multiple smaller greenhouses could improve city-wide accessibility [128]. Both large- and small-scale innovation is proving successful across a diversity of challenges in the industry. Ecuador's use of Geographic Information Systems (GIS) and mapping technologies to assess suitability for rooftop greenhouses has proven successful in identifying the optimal locations for houses [130]. In Germany, the first supermarket with a greenhouse roof was recently built, which utilizes an aquaponic system, using fish biowaste to fertilize the plants, serving as a biofilter to clean the water and return it to the fish tank [131]. Another notable aquaponic urban greenhouse exists in Sweden, where tropical commodities, including bananas and mangoes, are being locally grown [122]. The use of regionally advantageous technological developments is driving greenhouse operations for food sustainability.

Technological application is still challenging in some regions due to cost–benefit concerns. Greenhouse implementation in rural and impoverished areas around the globe can be less financially accessible as they do not reap the benefits of industrially integrated operations. For example, initiatives to help address food insecurity in mountainous regions of India and Nepal are challenging due to a lack of financial feasibility [132]. To meet these unique challenges, locally sourced building resources are being leveraged, and in Nepal's Humla district, solar greenhouses are designed for reduced heat loss and improved performance while being constructed from locally sourced wood, rocks, and mud [132].

The financial challenges of expanding operations can be limiting to implementation of automation in operations and government and NGO incentives can assist. In the rural far-northern regions of Russia, the potential for vertical farming using hydroponic systems to locally supply produce has been supported by significant federal financial incentives [133,134]. With remotely controlled climate control systems becoming more common in industrial greenhouse operations, advancements in AI, machine learning, and robotics have been identified as critical in order to meet sustainability goals, overcome labour shortages and pest challenges, and increase revenues [122,126]. Iceland's computerized geothermal climate control systems supporting automated hydroponic and soilless greenhouses have been successful, though they still depend on government funding and tariffs to compete economically with imported vegetables [123].

In addition to automation, the implementation of more efficient lighting systems has also been identified as critical in northern greenhouses [126]. Currently in Sweden, artificial lighting is seen as cost prohibitive so many operations use only natural lighting [122]. To address these concerns, neighbouring Norway has conducted research to optimize the lighting parameters in conjunction with ventilation [135]. With continued developments in anti-reflective, light diffusing, anti-fog, and solar cell integrated greenhouse glass, the challenge of sufficient light is being addressed [136]. While not in a northern climate, efforts to adapt the light itself to optimize the efficiency of greenhouse operations are also taking place, which could benefit northern greenhouse operations. In Australia, pink and orange plastics designed for retrofitting existing greenhouse operations are being used to obtain optimal frequencies of light for accelerating plant growth [137].

While northern greenhouse advancements are directly related to implementation in Canadian operations, all greenhouse technological advancements should be considered for potential utilization to further Canadian operations. Advancements to combat issues of arable land in the Caribbean and to produce crops not suited to the warmer climate, such as lettuces and berries, may hold lessons for Canadian operations [138]. The longevity of the greenhouses themselves is a concern to many regions with harsh conditions, including Canada. Adaptations to limit hurricane impacts and earthquakes could be applied for some of Canada's extreme weather events. With global warming trends and extreme weather events becoming increasingly unpredictable, technological advancements from all regions will be important to consider for innovating greenhouse programs [137,139].

## 5. Conclusions

Climate change and global population growth are straining global food production systems. Global responses to the COVID-19 pandemic, including the shutdown of international trade and human travel, highlight a need to bolster Canadian food security. It has been predicted that globalization will continue to decrease after the pandemic, invariably impacting food supply chains [140]. Facilitating a robust domestic food supply and decreasing reliance on international supply chains could promote self-reliance and food security for Canadians. Improving the stability and performance of the Canadian greenhouse industry is one way to increase resiliency to supply shocks due to international trade restrictions, thus improving crisis preparedness [141].

Certain opportunities for growth in the Canadian greenhouse industry presented here are achievable in the short-term, such as increasing the use of molecular biosurveillance tools to reduce crop losses. Some scenarios are longer-term, capital-intensive, and could impact the entire Canadian agriculture system. One notable obstacle facing the expansion of the greenhouse industry is the large capital investment required to develop indoor growing systems as compared to traditional agriculture [1] (pp. 483–498). Achieving the recommendations outlined here requires government initiatives with the integration of research institutions, trade associations, and the involvement of private stakeholder investment. By developing a comprehensive and actionable plan with all stakeholders to expand greenhouse vegetable production, national food security could be enhanced and protected into the future. The

Canadian greenhouse industry has the potential to be a key component of Canada's growth towards a more self-reliant food production system.

**Author Contributions:** Conceptualization, R.G.Y., S.A., G.L., J.M.K.; investigation, S.A., G.L., R.G.Y., J.M.K.; writing—original draft preparation, S.A., G.L., R.G.Y., J.M.K.; writing—review and editing, S.A., G.L., R.G.Y., J.M.K., E.J.C., N.B., R.H.H.; supervision, R.G.Y., J.M.K., E.J.C., R.H.H.; project administration, R.G.Y., J.M.K., R.H.H.; funding acquisition, R.G.Y., R.H.H. All authors have read and agreed to the published version of the manuscript.

**Funding:** This research was funded in part by the Ontario Ministry of Agriculture, Food and Rural Affairs (OMAFRA), through the Ontario Agri-Food Innovation Alliance and the Leading to the Accelerated Adoption of Innovative Research (LAAIR) award.

**Data Availability Statement:** Data sharing is not applicable to this article as no datasets were generated or analysed during the current study.

**Acknowledgments:** Thank you to Justine Taylor of the OGVG for reviewing this manuscript and providing valuable comments. Thank you to Aryana Rabii, Julian Nino, and Jacob Insley for their initial contributions during the planning stages of this manuscript. Thank you to Ashley Chen and Reese Solomon for assistance editing and formatting this manuscript. This is a contribution to the University of Guelph's Food from Thought initiative supported by the Canada First Research Excellence Fund.

**Conflicts of Interest:** The authors declare no conflict of interest.

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
