# Peer review of "Canadian Greenhouse Operations and Their Potential to Enhance Domestic Food Security"

_agronomy, doi:10.3390/agronomy11061229_

Round 1
Reviewer 1 Report
The manuscript agronomy-1244923, provides interesting information on the current state of Canadian greenhouse industry and the perspectives to improve their operations. The review is well-written and discussed. The challenge for modern agriculture and future prospects are well presented.
My only concer is: Line 380. What do you mean by 'Venlo structural materials'?
Author Response
Thank you very much for your thoughtful review. Your comment is well received and we have clarified the section of writing you were concerned about and changed the writing to the following that can be found starting on line 389.
Determining the style of greenhouse is an important factor to consider when balancing energy costs. Dutch Venlo greenhouse structures, most commonly used in the Netherlands and Australia, maximize carbon dioxide enrichment, utilize efficient cooling and ventilation systems, and use horizontal screens to reduce heat demand and energy costs [105,111-113].
Thank you for your consideration and time.

Reviewer 2 Report
The manuscript “Canadian greenhouse operations and their potential to enhance domestic food security” analyses the Canadian greenhouse industry and its position against possible food supply shortages.
After the analysis of the size and structure of the sector, the manuscript describes the strength and weaknesses of international trade. The paragraph describing the challenges of the Canadian sector is interesting and may provide some useful hints for improving state self-sufficiency. Moreover, the paragraph dealing with the opportunities of the sector provides an accurate analysis of the economic and technical approaches which may help the sector.
The topics presented are interesting, and this kind of review is still missing.
I still feel a Discussion is needed. Involving the comparison with other countries and the solution they have adopted.
Author Response
Thank you very much for your thoughtful comments. We have taken your suggestion and provided a discussion section with respect to Canadian greenhouse operations and innovation and how other world jurisdictions are approaching the problems. We have included this discussion near the end of the paper right before the Conclusion section starting at line 430. Please find the additional text below...
- Global Greenhouse Development Initiatives
The global community has been, and will continue to be, impacted by the effects of both climate change and COVID-19 on food security. Many countries have recently put a heightened focus on domestic supply, including greenhouse vegetable cultivation [119,120]. In Singapore, for example, border closures spurred efforts to realize the country’s “30 by 30” goal of 30% of food needs met domestically by 2030, resulting in increased investment in high-tech greenhouse development [120].
There are added challenges for more northern countries with harsher climates, like Canada, to increase greenhouse vegetable production. Finland, Sweden, Norway, Iceland, and Russia are looking to improve upon their existing greenhouse industries, which, like Canada, also commonly grow cucumbers, tomatoes, and peppers [121-124]. In fact, the Russian cucumber market is already almost 100% supplied by domestic greenhouse production [125]. In these northern regions, the major challenge for greenhouse production is high energy requirements for year-round heating [126]. Innovative solutions are continually being developed. In Sweden, over the past decade, wood chip boilers have largely replaced fossil fuels to become the major greenhouse heating method [122]. Some operations use waste heat from nearby factories or biogas generators from adjacent farms, and implementation of confluent air jet climate control systems is being explored commercially [122]. Research has also indicated the potential for heating greenhouses using manure-based biogas where it is expected that up to 11% of the Swiss tomato demand could be supplied domestically in this manner [127]. Icelandic greenhouse operations have been reported to harness geothermal heat to extend their growing season [123].
In urban areas, the problem of heating can be addressed by developing rooftop greenhouses coupled to the buildings below or nearby. In a dually beneficial manner, rooftop greenhouses provide insulation which prevents heat loss in the winter and absorbs excess heat in the summer, preventing detrimental urban heat island effects [128]. In addition to heat waste, urban greenhouses can be coupled to industrial or residential buildings to make use of carbon dioxide outputs [128,129]. Countries across the Americas and Europe have developed large-scale rooftop greenhouses [128,129,130,131], and analysis has indicated that in addition to these larger efforts that multiple smaller greenhouses could improve city-wide accessibility [129]. Both large and small-scale innovation is proving successful across a diversity of challenges in the industry. Ecuador’s use of Geographic Information System’s (GIS) and mapping technologies to assess suitability for rooftop greenhouses has proven successful in identifying optimal locations for houses [131]. In Germany, the first supermarket with a greenhouse roof was recently built which utilizes an aquaponic system, using fish biowaste to fertilize the plants, which serve as a biofilter to clean the water and return it to the fish tank [132]. Another notable aquaponic urban greenhouse exists in Sweden, where tropical commodities, including bananas and mangoes, are being locally grown [122]. The use of regionally advantageous technological developments is driving greenhouse operations for food sustainability.
Technological application is still challenging in some regions due to cost benefit concerns. Greenhouse implementation in rural and impoverished areas around the globe can be less financially accessible as they do not reap the benefits of industrially integrated operations. For example, initiatives to help address food insecurity in mountainous regions of India and Nepal are challenging due to a lack of financial feasibility [133]. To meet these unique challenges, locally sourced building resources are being leveraged, and in Nepal’s Humla district, solar greenhouses are designed for reduced heat loss and improved performance while being constructed from locally sourced wood, rocks, and mud [133].
The financial challenges of expanding operations can be limiting to implementation of automation in operations and government and NGO incentives can assist. In rural far-northern regions of Russia, the potential for vertical farming using hydroponic systems to locally supply produce has been supported by significant federal financial incentives [134,135]. With remotely controlled climate control systems becoming more common in industrial greenhouse operations advancements in AI, machine learning, and robotics have been identified as critical in order to meet sustainability goals, overcome labour shortages and pest challenges, and increase revenues [122,126]. Iceland’s computerized geothermal climate control systems supporting automated hydroponic and soilless greenhouses have been successful, though they still depend on government funding and tariffs to compete economically with imported vegetables [123].
In addition to automation, the implementation of more efficient lighting systems has also been identified as critical in northern greenhouses [126]. Currently in Sweden, artificial lighting is seen as cost prohibitive so many operations use only natural lighting [122]. To address these concerns, neighbouring Norway has conducted research to optimize lighting parameters in conjunction with ventilation [136]. With continued developments in anti-reflective, light diffusing, anti-fog, and solar cell integrated greenhouse glass, the challenge of sufficient light is being addressed [137]. While not in a northern climate, efforts to adapt the light itself to optimize the efficiency of greenhouse operations are also taking place that could benefit northern greenhouse operations. In Australia, pink and orange plastics designed for retrofitting existing greenhouse operations are being used to obtain optimal frequencies of light for accelerating plant growth [138].
While northern greenhouse advancements are directly related to implementation in Canadian operations, all greenhouse technological advancements should be considered for potential utilization to further Canadian operations. Advancements to combat issues of arable land in the Caribbean and to produce crops not suited to the warmer climate like lettuces and berries may hold lessons for Canadian operations [139]. The longevity of greenhouses themselves is a concern to many regions with harsh conditions, including Canada. Adaptations to limit hurricane impacts and earthquakes could be applied for some of Canada’s extreme weather events. With global warming trends and extreme weather events becoming increasingly unpredictable, technological advancements from all regions will be important to consider for innovating greenhouse programs [138,140].

This manuscript is a resubmission of an earlier submission. The following is a list of the peer review reports and author responses from that submission.
Round 1
Reviewer 1 Report
In my opinion, this review paper is judged to be at the level of a simple information report about greenhouse industry. It's also not an exact figure, but it's a fact that most researchers to study a greenhouse know.
To be accepted, for example, when a dynamic modeling technique is applied for the production of horticultural crops in a greenhouse, the productivity of crops is improved by certain factors, and these facts can substantially affect food security. Thus It is expected that more specific contents will be needed.
Reviewer 2 Report
Overall, agronomy-1138634 is nicely written review on Canadian greenhouse industry and its importance in ensuring the food security in Canada during the crisis caused by COVID-19 pandemic. The title properly reflects the subject of the paper. The introduction summarizes resent research related to the topic. The structure of the review is clear. The authors have used the adequate references.
Reviewer 3 Report
The paper reports the state of the Canadian greenhouse industry in a perspective of overcoming future food supply shortages. The review could be more articulated and extensive. However, the topics presented are interesting and represent a starting point for greenhouse industry expansion. Introduction: The authors state that the pandemic situation has highlighted the importance of regional self-reliance in food production. On the other hand, in the Conclusion there is a mention to climate change. In my opinion, the latter is a very interesting point which should be analysed. I don’t think the only mention in the Conclusion is useful and meaningful. State of the art “Size, Structure and Trends” and “International Trade” look like a summary of reference 7. I think the statement should be corroborated by additional references. I feel a Discussion is needed. Involving the comparison with other countries and the solution they’ve adopted. Here are some specific comments: Line 49: specify “Canadian dollar” Line 55-57: this is a repetition of the previous sentence